# Advance care planning in patients with advanced cancer: A 6-country, cluster-randomised clinical trial

Ida J. Korfage[1]*, Giulia Carreras[2], Caroline M. Arnfeldt Christensen[3,4], Pascalle Billekens[5], Louise Bramley[6], Linda Briggs[7], Francesco Bulli[2], Glenys Caswell[8], Branka Červ[9], Johannes J. M. van Delden[10], Luc Deliens[11], Lesley Dunleavy[12], Kim Eecloo[11], Giuseppe Gorini[2], Mogens Groenvold[3,4], Bud Hammes[7], Francesca Ingravallo[13], Lea J. Jabbarian[1], Marijke C. Kars[10], Hana Kodba-Čeh[9], Urska Lunder[9], Guido Miccinesi[2], Alenka Mimić[9], Polona Ozbič[9], Sheila A. Payne[12], Suzanne Polinder[1], Kristian Pollock[8], Nancy J. Preston[12], Jane Seymour[14], Anja Simonič[9], Anna Thit Johnsen[3,4], Alessandro Toccafondi[2], Mariëtte N. Verkissen[11], Andrew Wilcock[15], Marieke Zwakman[10], Agnes van der Heide[1‡], Judith A. C. Rietjens[1‡]

1 Department of Public Health, Erasmus MC, Rotterdam, Netherlands, 2 Clinical Epidemiology, Oncological Network, Prevention and Research Institute (ISPRO), Florence, Italy, 3 Department of Public Health, University of Copenhagen, Copenhagen, Denmark, 4 Department of Palliative Medicine, Research Unit, Bispebjerg Hospital, Copenhagen, Denmark, 5 Laurens, Rotterdam, Netherlands, 6 Institute of Nursing and Midwifery Care Excellence, Nottingham University Hospitals NHS Trust, Nottingham, United Kingdom, 7 Respecting Choices, C-TAC Innovations, Oregon, Wisconsin, United States of America, 8 School of Health Sciences, University of Nottingham, Nottingham, United Kingdom, 9 University Clinic of Respiratory and Allergic Diseases Golnik, Golnik, Slovenia, 10 Julius Centre for Health Sciences and Primary Care, UMC Utrecht, Utrecht, Netherlands, 11 End-of-Life Care Research Group, Vrije Universiteit Brussel and Ghent University, Brussels, Belgium, 12 International Observatory on End of Life Care, Division of Health Research, Lancaster University, Lancaster, United Kingdom, 13 Department of Medical and Surgical Sciences (DIMEC), University of Bologna, Bologna, Italy, 14 Health Sciences School, University of Sheffield, Sheffield, United Kingdom, 15 School of Medicine, University of Nottingham, Nottingham, United Kingdom

‡ These authors are joint senior authors on this work.
* i.korfage@erasmusmc.nl

**Data Availability Statement:** Data cannot be shared publicly because permission as provided by patients does not allow for that. Researchers can contact the Department of Public Health of

## Abstract

### Background

Advance care planning (ACP) supports individuals to define, discuss, and record goals and preferences for future medical treatment and care. Despite being internationally recommended, randomised clinical trials of ACP in patients with advanced cancer are scarce.

### Methods and findings

To test the implementation of ACP in patients with advanced cancer, we conducted a cluster-randomised trial in 23 hospitals across Belgium, Denmark, Italy, Netherlands, Slovenia, and United Kingdom in 2015–2018. Patients with advanced lung (stage III/IV) or colorectal (stage IV) cancer, WHO performance status 0–3, and at least 3 months life expectancy were eligible. The ACTION Respecting Choices ACP intervention as offered to patients in the intervention arm included scripted ACP conversations between patients, family members, and certified facilitators; standardised leaflets; and standardised advance directives.

Erasmus University Medical Center (secretariaat.mgz@erasmusmc.nl) to discuss potential limited use of the data.

**Funding:** AH declares grant funding for the submitted work from the European Union's Seventh Framework Programme FP7/2007-2013 under grant agreement n° 602541. The funders had no role in study design, data collection and analysis, decision to publish, or preparation of the manuscript.

**Competing interests:** I have read the journal's policy and the authors of this manuscript have the following competing interests: BH and LB are developers of Respecting Choices and report personal fees from Gundersen Health, outside the submitted work.

**Abbreviations:** ACP, advance care planning; AD, advance directive; EF10, emotional function 10-item short form; EORTC, European Organisation for Research and Treatment of Cancer; ICC, intraclass correlation coefficient; RC, Respecting Choices.

Control patients received care as usual. Main outcome measures were quality of life (operationalised as European Organisation for Research and Treatment of Cancer [EORTC] emotional functioning) and symptoms. Secondary outcomes were coping, patient satisfaction, shared decision-making, patient involvement in decision-making, inclusion of advance directives (ADs) in hospital files, and use of hospital care. In all, 1,117 patients were included (442 intervention; 675 control), and 809 (72%) completed the 12-week questionnaire. Patients' age ranged from 18 to 91 years, with a mean of 66; 39% were female. The mean number of ACP conversations per patient was 1.3. Fidelity was 86%. Sixteen percent of patients found ACP conversations distressing. Mean change in patients' quality of life did not differ between intervention and control groups (T-score −1.8 versus −0.8, $p = 0.59$), nor did changes in symptoms, coping, patient satisfaction, and shared decision-making. Specialist palliative care (37% versus 27%, $p = 0.002$) and AD inclusion in hospital files (10% versus 3%, $p < 0.001$) were more likely in the intervention group. A key limitation of the study is that recruitment rates were lower in intervention than in control hospitals.

## Conclusions

Our results show that quality of life effects were not different between patients who had ACP conversations and those who received usual care. The increased use of specialist palliative care and AD inclusion in hospital files of intervention patients is meaningful and requires further study. Our findings suggest that alternative approaches to support patient-centred end-of-life care in this population are needed.

## Trial registration

ISRCTN registry ISRCTN63110516.

## Author summary

### Why was this study done?

- Advance care planning (ACP) has been widely advocated as an approach to support patients, relatives, and healthcare professionals in reflecting on and discussing patients' preferences and to adapt care and treatment accordingly.

- There is little evidence of its effectiveness in relation to patients with advanced cancer in Europe.

### What did the researchers do and find?

- We conducted a study in 23 hospitals in 6 European countries, including 1,117 patients with advanced lung or colorectal cancer.

- Depending on the hospital where they were treated, they were offered ACP conversations with a certified facilitator or they were offered care as usual.

- Sixty-seven percent of patients considered the ACP conversations helpful, and most patients who took part in the ACP conversations appointed a relative who could represent their interests if they would not be able to do so themselves.

- Thirty-seven percent of patients in the intervention group completed a form to record their preferences for future care.

- We found that ACP conversations did not have an impact on patients' quality of life, coping, or involvement in decision-making processes; patients who had had ACP conversations more often received specialist palliative care.

**What do these findings mean?**

- The findings of the ACTION study did not provide evidence to support the use of a structured ACP intervention to improve the quality of life of patients affected by advanced lung or colorectal cancer.

- There is some evidence that patients taking part in ACP conversations were more likely to receive palliative care, and more likely to have their documented preferences recorded in their medical records.

- Further research is required to establish how patients can best be supported to formulate and, if they wish, to document their preferences for future care.

## Introduction

End-of-life discussions between patients, relatives, and healthcare professionals are associated with less burdensome interventions near death, earlier hospice referrals [1], improved emotional functioning [2,3], and better symptom resolution [3]. Still, timely and adequately providing patients and their families with an opportunity to prepare for the changes wrought by serious progressive illness and to explore patients' preferences is a challenge [4,5]. Advance care planning (ACP) enables individuals to define goals and preferences for future medical treatment and care, to discuss these goals and preferences with family and healthcare providers, and to record and review these preferences if appropriate [6]. ACP interventions have the potential to prepare patients for decision-making when they are unable to make their own decisions. ACP interventions typically include 1 or more focused, personal conversations between patients and healthcare professionals about patients' personal values, life goals, and preferences regarding future medical treatment and care. Such conversations have been reported to reduce hospital admissions at the end of life, to increase compliance of provided care with patients' wishes, and to increase satisfaction with care among older people and nursing home residents [7].

While ACP is a promising approach to improve the quality of life of patients with advanced cancer [7,8], evidence on its effectiveness for this patient group is limited [9]. We updated a 2019 systematic review of randomised controlled trials (RCTs) about ACP for patients with advanced cancer [10] and identified 6 RCTs of complex ACP interventions for advanced cancer patients: 1 conducted in the UK ($n$ = 77) [11], 2 in Australia ($n$ = 120–208) [12,13], and 3 in the US ($n$ = 155–278) [14–16]. In these studies, ACP was found to increase the proportion of patients engaging in conversations about future medical treatment and care, but not to

affect satisfaction with healthcare. No effects on place of death and treatment received at the end of life were found, while conflicting results were reported about the completion of advance directives (ADs) and patients' quality of life. All RCTs were at high risk of performance bias, attrition bias, or other kinds of bias. We therefore performed a large-scale RCT in 6 European countries to evaluate the effects of a complex ACP intervention on the quality of life, operationalised as emotional functioning, and symptoms of patients with advanced lung or colorectal cancer. Secondary outcomes were coping, patient satisfaction, shared decision-making, patient involvement in decision-making, AD inclusion in hospital files, and use of hospital care.

## Methods

### Study design

ACTION is a multicentre cluster-randomised controlled trial carried out in 23 hospitals in 6 European countries (Belgium, Denmark, Italy, the Netherlands, Slovenia, and the United Kingdom). We opted for cluster randomisation to prevent contamination. Using a computer-based generator tool, per country and per pair of comparable hospitals (academic or non-academic), hospitals were randomised by the study coordinator to the intervention arm, providing usual care and ACP, or the control arm, providing usual care. Per hospital usually 2 departments participated, e.g., a pulmonology and an oncology department. All patients with advanced lung (stage III or IV) or colorectal cancer (stage IV), WHO performance status 0–3, an estimated life expectancy of at least 3 months, and competence to give consent were eligible (see S1 Text). When a care team considered patients eligible, they were asked to consider participation in ACTION. Patients who wanted to consider participation were contacted by the researcher team and provided with more information about the study. Patients in the intervention hospitals received information about the intervention. Those in control hospitals were informed that ACTION focused on preparing patients for decision-making about care, and that they would receive usual care. Patients were given unrestricted time to consider participation and were informed that they were free to withdraw from participating in the study without any effect on their care. Patients who provided written informed consent were included and followed until 12 months after inclusion. The trial procedures have been described in detail in a protocol paper [17]. The trial was registered in the ISRCTN registry (ISRCTN63110516) as of 10 March 2014. We report the study according to CONSORT reporting guidelines.

### The intervention

Respecting Choices (RC) is a comprehensive, structured ACP programme that was developed in La Crosse, Wisconsin, in the US (https://respectingchoices.org/). RC was successfully trialled among older people in Australia [18]. We developed and evaluated the ACTION RC ACP intervention. This was an adapted and integrated version of 2 of the 3 stages of the RC facilitated ACP conversations (First Steps and Advanced Steps). The ACTION RC ACP intervention includes 3 components (see S2 Text): (1) facilitated structured ACP conversations, (2) the My Preferences form, and (3) information leaflets.

**Facilitated structured ACP conversations.** Healthcare professionals who were certified to deliver the ACTION RC ACP intervention used scripted conversation guides to support patients and their relatives in exploring their understanding of the illness; reflecting on their goals, values, and beliefs; and discussing their preferences for future treatment and care. Depending on the choice of the patient, the intervention involved 1 or 2 conversations, with or without a personal representative being present. Conversations took place in the hospital or at patients' homes.

**My Preferences form.**   The My Preferences form is a study-specific form where patients can document their preferences (see S3 Text). Depending on local regulations, the My Preferences form may be considered as a formal AD or an informal expression of wishes. It was developed in an iterative process with input from stakeholders, including patient representatives, clinicians, and researchers from all participating countries. It consists of open sections regarding 'living well', 'worries and fears', 'beliefs', and 'hopes', and structured sections to indicate preferences regarding cardiopulmonary resuscitation (CPR), goals of future care, and final place of care. Patients were offered the option of completing a My Preferences form, either during or after the ACP conversation. They were advised that they could provide copies to their family and healthcare professionals and could adapt their My Preferences form if needed.

**Information leaflets.**   Information leaflets regarding ACP and the role of the personal representative were provided to all patients who participated in the intervention. Where relevant, patients were also provided with information leaflets about CPR, artificial ventilation, and artificial feeding.

Cross-cultural adaption of intervention and training materials was required to make them appropriate for the countries in this study whilst maintaining the essentials of the content, structure, and integrity of the original intervention. Materials were tested in a feasibility study with 53 patients, 18 relatives, and 29 healthcare professionals and subsequently finalised. Aiming for maximum uniformity in the delivery of the intervention across the 6 countries, all ACTION RC instructors followed the RC First and Advanced Steps training programme together. In each country, an ACTION RC instructor provided facilitators with a 2-day competency-based training programme, in the local language. In total, 39 facilitators, predominantly nurses, were certified.

Per facilitator, fidelity checks were conducted twice, once halfway through the inclusion period and once towards the end of the inclusion period. ACP conversations were audio-recorded, and facilitators' compliance with the conversation intervention was systematically evaluated by local ACTION RC instructors, using a pilot-tested fidelity checklist that covered the key elements of the conversation guides.

## Outcome measures

At baseline and at 11–12 weeks (follow-up assessment 1) and 19–20 weeks (follow-up assessment 2) after inclusion, patients completed a written questionnaire. Quality of life, assessed with 10 items of the European Organisation for Research and Treatment of Cancer (EORTC) emotional functioning item bank [19–21] and symptoms (EORTC QLQ-C15-PAL [22]) were the primary outcomes. Secondary outcomes were coping (COPE, Brief COPE [23–25]), satisfaction with care (items of the EORTC IN-PATSAT [26]), satisfaction with the intervention (9 study-constructed items), shared decision-making (Assessment of Patients' Experience of Cancer Care [APECC] decision-making self-efficacy scale [27]), and patient involvement in medical decision-making (4 study-constructed items). The availability of completed ADs in the hospital medical file and use of medical care were extracted from medical files in the hospital where patients were recruited. A medical file checklist was developed to allow structured and uniform data collection at 1 year post-inclusion. Due to closure of the data collection, medical files were not checked for patients included after 30 April 2017.

## Statistical analysis

We aimed at an overall power of 0.9 (alpha 0.05) to detect a difference between the intervention and control group at the first follow-up assessment of at least 0.5 standard deviation on

the 4-item emotional functioning scale (EF4) of the EORTC-QLQ-C30, assuming an intraclass correlation coefficient (ICC) of 0.1 [17]. We expected to include patients in at least 10 intervention and 10 control hospitals (40 departments). We anticipated around 15% of included patients would die before the first follow-up assessment and about 10% would drop out, resulting in an expected attrition rate of 25%. We therefore needed to include 68 participants per hospital, resulting in 1,360 patients [17]. Originally, 22 hospitals were randomised. For logistic reasons, no patients were included in 1 hospital. During the study, 2 additional hospitals were randomised to increase the inclusion rate, resulting in a total number of 23 hospitals.

When during data collection the EORTC emotional function 10-item short form (EF10) appeared to have better precision and relative validity than EF4 [28], we opted for EF10 as our primary outcome measure; the new target number of patients while maintaining the same power was 1,088.

Statistical analyses were by intention-to-treat. Patients lost to follow-up and other missing information were taken into account by performing a multiple imputation (MI) procedure [29]. The MI procedure was implemented using the Multivariate Imputation by Chained Equations (MICE) algorithm with $M$ = 37 imputations [30]. At the end of the MICE procedure we analysed each of the $M$ complete datasets applying standard statistical methods and we combined the results according to Rubin's rule [29] (see Box A, Table A, and Table B in S4 Text). We compared differences in change scores between the 2 arms and evaluated the association between treatment and each outcome measure using a multilevel regression model with random intercept adjusted for the baseline value of the measure, allowing us to take into account the intraclass correlation occurring between patients attending the same hospital.

Assuming that the RC ACP intervention would not have influenced survival, we planned to perform the analyses on the subset of patients who survived until the first and second follow-up assessments. Moreover, we checked for determinants of emotional functioning apart from the intervention and for an interaction effect of the intervention and country using a multilevel regression model that took into account clustering at the hospital level. The model was adjusted for the baseline level of the measure, country, and individual characteristics (i.e., sex, having children, age, years of education, living with a spouse/partner, living place, religiosity, WHO performance status, current treatment, and cancer stage and type) selected on the basis of the Akaike information criterion for each imputed dataset [31]. Differences were considered significant if $p < 0.05$ (see Table C in S4 Text).

## Ethics

Ethical approval was obtained from research ethics committees in the coordinating centre at Erasmus MC (NL50012.078.14, v02) and in all participating countries. An international data and safety monitoring board monitored the trial and conducted 4 interim analyses during data collection.

## Results

### Participation and feasibility

Between 21 May 2015 and 6 February 2018, 3,748 patients were considered eligible, 2,748 (73%) were asked to participate, and 1,135 provided consent to participate. Of these, 5 withdrew their consent. The recruitment rate was 29% in the intervention group (445/1,523) versus 56% in the control group (685/1,225). Thirteen patients did not complete any questionnaires. We included data of 1,117 patients in the analyses (442 in the intervention group and 675 in the control group; see Fig 1). Patients' mean age was 66 years. In both groups, the majority of patients received systemic anti-cancer treatment at the time of study inclusion (Table 1).

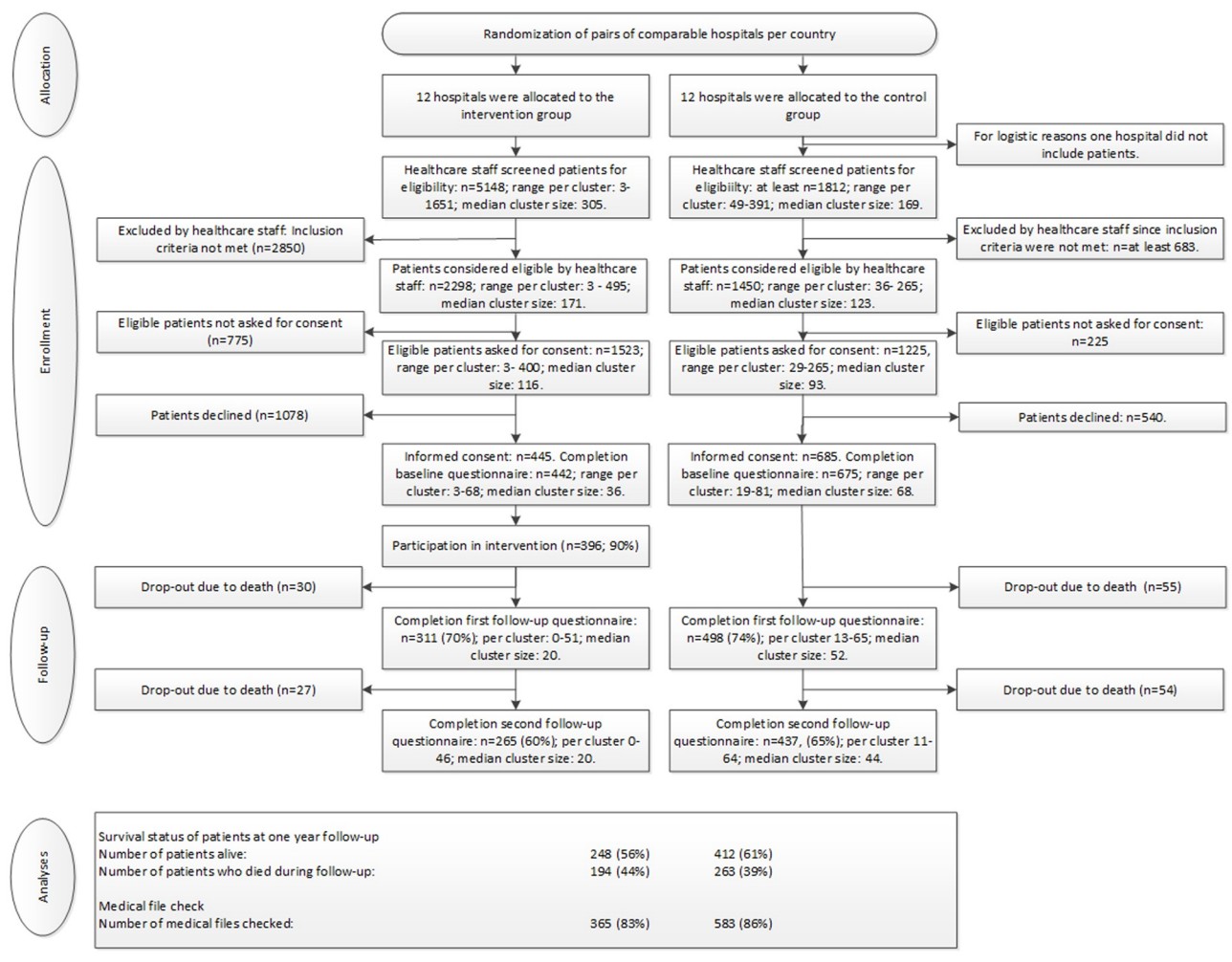

**Fig 1. Participant flowchart.**

During the 1-year follow-up period, 194 patients in the intervention group (44%) and 263 (39%) in the control group died: 30 patients in the intervention group (7%) and 55 patients in the control group (8%) died before follow-up assessment 1, whereas 57 (13%) and 109 (16%) patients, respectively, died before follow-up assessment 2.

## Intervention delivery

In the intervention group, 396 patients participated in ACP conversations (90%). Patients had on average 1.3 ACP conversations, which had a mean length of 93 minutes. Fidelity analyses showed that on average 86% of the key elements of the intervention were discussed.

## Primary and secondary outcome measures

Since the assumption that the RC ACP intervention did not influence survival was satisfied ($p = 0.57$ and $p = 0.41$ at follow-up assessments 1 and 2, respectively), we performed the analyses on the subset of patients who survived until the first and second follow-up assessments (92% and 85% of the sample, respectively). Of patients who survived, 22% and 26% were lost to follow-up at the first and second follow-up assessments, respectively. We found no

**Table 1. Sociodemographic and clinical characteristics of ACTION participants.**

| Characteristic | Intervention group (*n* = 442) | Control group (*n* = 675) |
|---|---|---|
| **Sociodemographic characteristics** | | |
| Years of education, mean (SD) | 13.1 (4.5) | 12.9 (4.7) |
| *Missing* | *56* | *94* |
| Female sex, *n* (%) | 173 (39) | 268 (40) |
| Living with a spouse/partner, *n* (%) | 303 (69) | 497 (74) |
| *Missing* | *10* | *17* |
| Having children, *n* (%) | 376 (85) | 583 (86) |
| *Missing* | *8* | *7* |
| Religiosity, *n* (%) | | |
| Religious | 207 (47) | 341 (51) |
| Not religious | 174 (39) | 228 (34) |
| Prefers not to specify | 51 (12) | 93 (14) |
| *Missing* | *10* | *13* |
| Considering oneself member of minority group, *n* (%) | 3 (1) | 7 (1) |
| *Missing* | *18* | *26* |
| Country of residence, *n* (%) | | |
| Belgium | 72 (16) | 135 (20) |
| Denmark | 68 (15) | 68 (10) |
| Italy | 31 (7) | 139 (21) |
| Netherlands | 84 (19) | 168 (25) |
| Slovenia | 72 (16) | 25 (4) |
| United Kingdom | 115 (26) | 140 (21) |
| **Clinical characteristic** | | |
| Diagnosis, *n* (%) | | |
| Lung cancer, stage III or IV | 271 (62) | 339 (50) |
| Colorectal cancer, stage IV | 171 (38) | 336 (50) |
| Years since diagnosis, mean (SD) | 1.2 (1.7) | 1.7 (2.4) |
| Range | 0.1–11.5 | 0.1–33.3 |
| *Missing* | *7* | *1* |
| Years since diagnosis of current stage, mean (SD) | 0.6 (0.9) | 1.0 (1.4) |
| Range | 0–6 | 0–11 |
| *Missing* | *2* | *3* |
| Receiving systemic treatment[1], *n* (%) | 349 (79) | 595 (89) |
| *Missing* | *2* | *3* |
| WHO performance status[2], *n* (%) | | |
| 3 In bed/ sitting for more than half of the day | 10 (2) | 8 (1) |
| 2 Up for more than half of the day | 74 (17) | 55 (8) |
| 1 No heavy psychical work | 243 (55) | 343 (51) |
| 0 Fully active | 109 (25) | 261 (39) |
| *Missing* | *6* | *8* |

[1]Includes chemotherapy, immunotherapy, and targeted therapy.

statistically significant differences between the intervention and control group in change in EORTC emotional functioning score (EF10) at the first (−1.8 versus −0.8, *p* = 0.59; Table 2) or second (−2.3 versus −0.2, *p* = 0.10) follow-up assessment. Further analyses showed that there was no interaction effect of intervention and country either (*p* = 0.41).

**Table 2. ICC and changes in outcome scores between follow-up assessment 1 and baseline and follow-up assessment 2 and baseline (with 95% confidence intervals), with *p*-values from multilevel regression model with random intercept and with hospital as random level adjusting for baseline levels of each endpoint.**

| Outcome | Follow-up assessment 1 versus baseline | | | | Follow-up assessment 2 versus baseline | | | |
|---|---|---|---|---|---|---|---|---|
| | ICC | Intervention group (*n* = 412) | Control group (*n* = 620) | *p*-Value | ICC | Intervention group (*n* = 385) | Control group (*n* = 566) | *p*-Value |
| **Emotional functioning (EORTC EF10)*** | 0.030 | −1.8 (−4.2, 0.7) | −0.8 (−1.5, −0.1) | 0.59 | 0.001 | −2.3 (−5.8, 1.2) | −0.2 (−1.4, 0.9) | 0.10 |
| **Quality of life and symptoms (EORTC QLQ-C15-PAL)^** | | | | | | | | |
| Overall quality of life | 0.053 | −5.7 (−14.2, 2.9) | −2.0 (−4.0, 0.1) | 0.22 | 0.026 | −5.4 (−13.8, 2.9) | −1.4 (−3.6, 0.8) | 0.23 |
| Emotional functioning | 0.030 | −3.6 (−8.2, 1.1) | −1.7 (−3.4, 0.1) | 0.90 | 0.003 | −5.0 (−12.5, 2.5) | −3.3 (−7.5, 1.0) | 0.95 |
| Physical functioning | 0.049 | −6.5 (−16.3, 3.3) | −5.6 (−14.0, 2.9) | 0.56 | 0.022 | −8.8 (−23.1, 5.4) | −5.5 (−13.8, 2.8) | 0.06 |
| Pain | 0.015 | 6.0 (−9.7, 21.6) | 4.3 (−7.4, 16.0) | 0.45 | 0.019 | 8.4 (−12.4, 29.1) | 3.9 (−6.9, 14.7) | 0.05 |
| Dyspnoea | 0.045 | 5.4 (−9.7, 20.5) | 4.5 (−7.9, 16.8) | 0.53 | 0.030 | 5.8 (−10.5, 22.1) | 5.5 (−9.2, 20.2) | 0.63 |
| Insomnia | 0.031 | −0.9 (−4.5, 2.7) | 1.0 (−4.5, 6.5) | 0.29 | 0.034 | 2.6 (−7.5, 12.7) | −0.3 (−4.2, 3.6) | 0.42 |
| Appetite loss | 0.036 | 5.2 (−9.7, 20.2) | 3.8 (−7.3, 14.9) | 0.69 | 0.023 | 8.6 (−13.4, 30.6) | 5.9 (−9.8, 21.5) | 0.38 |
| Constipation | 0.024 | 1.2 (−5.8, 8.2) | 2.2 (−5.6, 10.0) | 0.69 | 0.023 | 1.0 (−5.8, 7.7) | 3.5 (−7.0, 14.1) | 0.75 |
| Fatigue | 0.026 | 5.3 (−9.1, 19.7) | 2.8 (−6.0, 11.5) | 0.36 | 0.025 | 7.0 (−11.1, 25.1) | 3.5 (−6.8, 13.8) | 0.15 |
| Nausea/vomiting | 0.010 | 2.7 (−6.0, 11.4) | 2.0 (−4.8, 8.9) | 0.88 | 0.010 | 3.2 (−6.6, 13.0) | 3.4 (−6.1, 12.8) | 0.99 |
| **Shared decision-making (APECC decision-making self-efficacy scale)^** | 0.098 | −3.3 (−7.3, 0.7) | −1.8 (−3.6, 0) | 0.77 | 0.095 | −2.5 (−5.0, 0.1) | −3.2 (−6.9, 0.6) | 0.24 |
| **Patient involvement in medical decision-making^** | 0.022 | 1.7 (−4.5, 7.8) | −1.9 (−3.9, 0.2) | 0.33 | 0.026 | 2.5 (−5.4, 10.3) | −1.9 (−4.2, 0.4) | 0.33 |
| **Coping (COPE)^** | | | | | | | | |
| Denial | 0.040 | 0.2 (−4.4, 4.8) | 2.6 (−5.6, 10.7) | 0.63 | 0.070 | 1.0 (−5.1, 7.0) | 2.8 (−6.0, 11.5) | 0.75 |
| Acceptance | 0.067 | −3.7 (−8.2, 0.9) | −3.0 (−6.6, 0.6) | 0.58 | 0.040 | −5.7 (−14.1, 2.7) | −3.2 (−7.2, 0.8) | 0.99 |
| Problem focused | 0.079 | −4.8 (−11.0, 1.5) | −3.7 (−8.6, 1.2) | 0.76 | 0.073 | −5.0 (−11.5, 1.6) | −3.6 (−8.0, 0.8) | 0.78 |
| **Satisfaction with care (EORTC IN-PATSAT)^** | | | | | | | | |
| Information provision by doctors | 0.055 | −3.3 (−6.8, 0.2) | −3.3 (−7.4, 0.9) | 0.59 | 0.060 | −2.6 (−5.1, −0.1) | −5.4 (−13.7, 2.9) | 0.57 |
| Information provision by nurses | 0.046 | −3.3 (−7.1, 0.5) | −3.9 (−9.0, 1.3) | 0.56 | 0.047 | −3.3 (−7.1, 0.6) | −5.6 (−14.1, 3.0) | 0.23 |
| General rating of received care | 0.067 | −3.9 (−9.1, 1.3) | −4.5 (−11.2, 2.1) | 0.54 | 0.054 | −5.0 (−12.5, 2.4) | −6.3 (−16.4, 3.8) | 0.45 |

The analyses are performed on imputed data (*M* = 37 imputations) on patients who survived to follow-up assessment 1 (*N* = 1,032; 85 deaths before follow-up assessment 1) for analyses on follow-up assessment 1 and on patients who survived to follow-up assessment 2 (*N* = 951; 166 deaths before follow-up assessment 2) for analyses on follow-up assessment 2.

*The score is transformed to a T-score metric with a general population mean of 50 (standard deviation 10).

^Possible score ranges from 0 to 100.

APECC, Assessment of Patients' Experience of Cancer Care; EF10, emotional function 10-item short form; EORTC, European Organisation for Research and Treatment of Cancer; ICC, intraclass correlation coefficient.

Change scores did not differ between arms for the EORTC QLQ-C15-PAL scales, coping, satisfaction with care, patient involvement in decision-making, or shared decision-making (Table 2). In the intervention group, 147 of 396 (37%) patients who had ACP conversations provided their facilitators with a copy of their completed My Preferences form. It is not known how many others completed the My Preferences form or another document indicating their preferences.

We analysed the medical files of 365 patients (83%) in the intervention group and of 583 control patients (86%). At 12 months post-inclusion, 37 medical files (10%) of patients in the intervention group contained ADs versus 15 in the control group (3%; *p* < 0.001). Indications that personal representatives were appointed were more often found in medical files of patients in the intervention group (33 versus 7 times, *p* < 0.001). During the 12 months of follow-up, 61% of patients in the intervention group and 56% of patients in the control group were

**Table 3. The documentation of preferences in medical files and the use of hospital care.**

| Outcome | Intervention group (*n* = 442) | Control group (*n* = 675) | *p*-Value |
|---|---|---|---|
| **AD in medical file** | | | |
| Completed AD in medical file, *n* (%) | 37 (10) | 15 (3) | <0.001* |
| *Missing* | *77* | *92* | |
| Type of AD, *n* (%) | | | |
| My Preferences form | 31 | Not applicable | |
| Other AD | 8 (2) | 15 (3) | 0.71* |
| Appointment of personal representative, *n* (%) | 33 (28 MPF, 4 other, 1 both) (94) | 7 (47) | <0.001* |
| *Missing* | *2* | *0* | |
| **Hospital care** | | | |
| Any hospitalisation, *n* (%) | 222 (61) | 328 (56) | 0.17* |
| *Missing* | *77* | *92* | |
| If any hospitalisation, number of days, mean (SD, range) | 15 (13, 0–63) | 14 (13, 0–75) | 0.53^ |
| Use of specialist palliative care, *n* (%) | 134 (37) | 160 (27) | 0.002* |
| *Missing* | *78* | *92* | |

*Chi-squared test on observed values.

^Unpaired *t* test.

AD, advance directive; MPF, My Preferences form.

hospitalised (*p* = 0.14). Their average number of inpatient hospital days was 15 and 14, respectively (*p* = 0.5; Table 3). In the intervention group, relatively more patients had used specialist palliative care services (*n* = 134 [37%] versus *n* = 160 [27%]; *p* = 0.002; Table 3).

Sixty-seven percent of patients considered the ACP conversations 'quite or very helpful', and 16% considered them 'quite or very stressful' (Table 4). Three serious adverse events (SAEs) related to the intervention were reported: 1 patient was distressed after reading the study materials and 2 after an ACP conversation. These SAEs were resolved.

## Discussion

We performed a large randomised controlled trial evaluating the effects of ACP. In 23 hospitals across 6 countries, we included 1,117 patients with advanced lung or colorectal cancer. Patients in the intervention group had ACP conversations with certified facilitators. We did not find any difference in effect on patients' quality of life, symptoms, coping, satisfaction with care, or shared decision-making at 11–12 weeks post-inclusion, nor did we find an effect on hospital admissions during 1 year of follow-up compared to the control group. Patients in the intervention group used specialised palliative care more often. Hospital files of patients in the intervention group contained ADs, and indications of appointed personal representatives as part of the ADs, more often than those of patients in the control group.

In both the intervention and the control arm, scores for shared decision-making, information as provided by doctors or nurses, and the rating of care as received declined from baseline. This unexpected finding could potentially be explained by increased awareness through study participation of the complexity of decision-making and information provision.

### Strengths and limitations

An important feature of our trial is its pragmatic nature, focusing on actual practice in countries with different healthcare systems and end-of-life care cultures, and varying degrees of familiarity with ACP, which increases the external validity and generalisability of our findings.

**Table 4. ACP process and its evaluation by patients.**

| ACP process outcome | Patients in intervention group (n = 442) |
|---|---|
| Patients who had ACP conversations; n (%) | 396 (90) |
| Number of conversations per patient; mean (range) | 1.3 (1–3) |
| Length of conversations in minutes; mean (SD, range) | 93 (43; 4–303) |
| One or more relatives attended ACP conversation; n (%) | |
| First conversation (n = 394) | 262 (67) |
| Second conversation (n = 116) | 95 (82) |
| Third conversation (n = 2) | 2 (100) |
| **Questions about the ACP process (n = 303)*** | |
| Evaluation of number of conversations; n (%) | |
| Too few | 30 (10) |
| Just right | 265 (89) |
| Too many | 3 (1) |
| Timing of ACP conversations; n (%) | |
| Too early | 48 (16) |
| Just right | 228 (76) |
| Too late | 25 (8) |
| ACP conversations considered to be helpful; n (%) | |
| Not at all | 18 (6) |
| A little | 82 (27) |
| Quite a bit | 119 (39) |
| Very much | 84 (28) |
| ACP conversations considered distressing; n (%) | |
| Not at all | 171 (56) |
| A little | 86 (28) |
| Quite a bit | 32 (11) |
| Very much | 14 (5) |

*Questions about the ACP process were included in follow-up questionnaires 1 and 2. If a participant answered the questions in both questionnaires, only the answers to follow-up questionnaire 2 were included.
ACP, advance care planning.

An additional strength is the high-quality research design in which we evaluated a uniform, multi-component ACP intervention that included structured, facilitated conversations, a uniform cross-cultural training programme, and fidelity evaluations. In addition, we developed the My Preferences form, aiming to enable participants to document their preferences in a format that was socially, culturally, legally, and ethically acceptable in all 6 participating countries. In a content analysis of 123 My Preferences forms, it was found that 43.9% of patients opted for comfort-focused care only; 75 preferred home as the final place of care, 20 preferred hospice, and 10 preferred hospital [32]. We included 1,117 patients, which is above the target number of 1,088 [28]. The ICC as observed for the primary outcome (EF10) turned out to be smaller than was assumed at the design stage, thus enhancing the trial power [33].

The study also has limitations. First, fewer eligible patients in the intervention group were asked to participate than in the control group (66% versus 84%), suggesting some level of gatekeeping [34]. In addition, recruitment rates were higher in control than in intervention hospitals. These factors may have resulted in unmeasured baseline differences between the study groups, which may also explain the fact that mortality rates were higher in the intervention than in the control arm. A systematic review of recruitment issues in palliative care

randomised controlled trials identified a number of barriers to recruitment that also appeared to be issues in this trial [34]. These include patients not being interested in the intervention, the burden of illness, and gatekeeping by healthcare professionals. We also know from the general literature that healthcare professionals struggle to introduce ACP and discuss end-of-life care issues. Further research is required to explore the complexity of recruitment in palliative care trials [35].

Second, information materials for the control groups also referred in some detail to the intervention. Although we support this from an ethical viewpoint, the information may have alerted patients in the control group to engage in decision-making processes and so may have reduced the contrast between the groups. Third, analyses were carried out on survivors only, whereas survivor average causal effect modelling or partly conditional inference could be a more sophisticated approach to address truncation by death [36,37]. However, in our data no significant differences were found in the proportion of patients who died between the intervention and control arm, and these proportions were small, especially at follow-up assessment 1. Fourth, the number of ACP conversations was limited to an average of 1.3 per patient. This may have affected the impact of the ACP programme. Finally, attrition of patients, although expected in this population, was rather high. Despite these limitations, the trial provides rigorous evidence of the effect of an internationally recognised programme. The results suggest both the challenges of timely discussion of preferences—and aligning the care of patients with advanced cancer accordingly—and the relevance of conducting randomised evaluation of interventions such as ACP programmes.

Several studies outside Europe have reported positive effects of comparable ACP programmes on various outcomes. For instance, the RC programme was found to increase satisfaction with care among older hospitalised patients in Australia [18]. RC facilitation improved the ACP knowledge and decreased the willingness to undergo life-sustaining treatments of ambulatory geriatric patients in the US [38]. However, in a recent study into the effects of facilitated ACP among frail older people in a Dutch population, no effects on quality of life were found [39].

The lack of comprehensive evidence of a positive impact of the complex ACP intervention evaluated in this study may be explained by (1) 1 or more of the characteristics of the intervention, (2) choice and timing of outcome measures, or (3) patients' preferences regarding ACP and ADs.

The first explanation relates to potential inadequacies of the intervention. We could not ensure that ACP documents were routinely completed, included in the medical notes, and acted upon by physicians. Also, the intervention was delivered in a research context, which required standardisation. As a result, the programme was not integrated with routine services, nor adapted to local circumstances and needs, which may have reduced its effect. For future research we would recommend exploring all options for broader involvement at the institutional level. Further, although patients were offered 2 conversations, the number of ACP conversations was limited to an average of 1.3 per patient. This may have affected the impact of the ACP programme.

The second potential explanation is that the choice and timing of outcome measurements may have been suboptimal. Our primary outcome measure was quality of life, operationalised by the EORTC emotional functioning items, at 11–12 weeks of follow-up. In 2017, a large international Delphi panel agreed that quality of life is not the most appropriate outcome of ACP [6]. Goal-concordant care might have been a better primary outcome, but is very difficult to measure because a validated measure to assess goal-concordant care is lacking [40]. Other studies have discussed the complexity of determining the right outcome measure of ACP studies as well [16,39,41]. Effects of ACP could also predominantly occur in the relational domain: patients and relatives may indeed have been supported to discuss and exchange views about

values, goals, and preferences, but this outcome was not explicitly assessed in our trial. The qualitative data as collected in the ACTION trial may shed some light on these potential effects. Although we checked medical files up until death for some patients, the timing of our outcome measurement may have been suboptimal: effects may occur at a later stage for at least some patients, at a time when their situation deteriorates and actual decisions have to be made.

The third potential explanation is that patients may prefer not to fully engage in ACP or make ADs. At the start of the study, the concept of ACP was almost unknown in Denmark [42], Italy, and Slovenia, and ADs had no legal status in Italy [17,43]. The lower recruitment rate in the intervention arm suggests that engaging in ACP in a study context was not attractive for all patients, the majority of whom received anti-cancer treatment. This is supported by 16% of participants reporting the ACP conversations to be distressing. Further, patients may feel reluctant to document specific preferences as these preferences may change [44], or patients may not consider documentation meaningful, either because they find it hard to envisage the future or because they trust their family or physician to decide what is best.

We conclude that in our large trial in European patients with advanced cancer, we did not find effects of the ACTION RC ACP intervention on quality of life, coping, patient satisfaction, or shared decision-making. Potential explanations relate to characteristics of the intervention, patients' preferences regarding ACP, and the choice of outcome measures. The increase in use of specialist palliative care and of inclusion of ADs in hospital files among patients who received the intervention is meaningful and requires further study. Our findings suggest that additional approaches to support patient-centred end-of-life care and to improve quality of life in this population are needed.

## Supporting information

**S1 CONSORT Checklist.**
(DOCX)

**S1 Text. Inclusion and exclusion criteria of the ACTION study.**
(DOCX)

**S2 Text. Description of the ACTION RC ACP intervention.**
(DOCX)

**S3 Text. The My Preferences form.**
(DOCX)

**S4 Text. Supporting box and tables.** Box: Handling of missing data. Table A: Number and proportion of missing values for sociodemographic and clinical variables, and for the questionnaire items used to build the scores. Table B: Distribution of loss to follow-up at follow-up assessments 1 and 2 by sociodemographic and clinical variables among surviving patients, with *p*-value from chi-squared test. Table C: Treatment effect on emotional functioning.
(DOCX)

## Acknowledgments

We gratefully acknowledge the contribution of Morten Petersen in EF10 calculations.

## Author Contributions

**Conceptualization:** Ida J. Korfage, Johannes J. M. van Delden, Mogens Groenvold, Jane Seymour, Agnes van der Heide, Judith A. C. Rietjens.

**Data curation:** Kim Eecloo, Lea J. Jabbarian.

**Formal analysis:** Ida J. Korfage, Giulia Carreras.

**Funding acquisition:** Ida J. Korfage, Johannes J. M. van Delden, Mogens Groenvold, Agnes van der Heide, Judith A. C. Rietjens.

**Investigation:** Ida J. Korfage, Caroline M. Arnfeldt Christensen, Louise Bramley, Francesco Bulli, Glenys Caswell, Branka Červ, Lesley Dunleavy, Kim Eecloo, Lea J. Jabbarian, Marijke C. Kars, Hana Kodba-Čeh, Urska Lunder, Guido Miccinesi, Alenka Mimić, Polona Ozbič, Sheila A. Payne, Kristian Pollock, Nancy J. Preston, Jane Seymour, Anja Simonič, Anna Thit Johnsen, Alessandro Toccafondi, Mariëtte N. Verkissen, Andrew Wilcock, Marieke Zwakman, Agnes van der Heide, Judith A. C. Rietjens.

**Methodology:** Ida J. Korfage, Giulia Carreras, Caroline M. Arnfeldt Christensen, Pascalle Billekens, Linda Briggs, Johannes J. M. van Delden, Luc Deliens, Giuseppe Gorini, Mogens Groenvold, Bud Hammes, Francesca Ingravallo, Urska Lunder, Guido Miccinesi, Sheila A. Payne, Suzanne Polinder, Nancy J. Preston, Anna Thit Johnsen, Agnes van der Heide, Judith A. C. Rietjens.

**Project administration:** Ida J. Korfage, Lesley Dunleavy, Kim Eecloo.

**Supervision:** Ida J. Korfage, Pascalle Billekens, Johannes J. M. van Delden, Bud Hammes, Suzanne Polinder, Kristian Pollock, Agnes van der Heide, Judith A. C. Rietjens.

**Validation:** Ida J. Korfage, Luc Deliens.

**Writing – original draft:** Ida J. Korfage, Giulia Carreras, Giuseppe Gorini, Mogens Groenvold, Bud Hammes, Francesca Ingravallo, Lea J. Jabbarian, Marijke C. Kars, Agnes van der Heide, Judith A. C. Rietjens.

**Writing – review & editing:** Ida J. Korfage, Giulia Carreras, Caroline M. Arnfeldt Christensen, Pascalle Billekens, Louise Bramley, Linda Briggs, Francesco Bulli, Glenys Caswell, Branka Červ, Johannes J. M. van Delden, Luc Deliens, Lesley Dunleavy, Kim Eecloo, Giuseppe Gorini, Mogens Groenvold, Bud Hammes, Francesca Ingravallo, Lea J. Jabbarian, Marijke C. Kars, Hana Kodba-Čeh, Urska Lunder, Guido Miccinesi, Alenka Mimić, Polona Ozbič, Sheila A. Payne, Suzanne Polinder, Kristian Pollock, Nancy J. Preston, Jane Seymour, Anja Simonič, Anna Thit Johnsen, Alessandro Toccafondi, Mariëtte N. Verkissen, Andrew Wilcock, Marieke Zwakman, Agnes van der Heide, Judith A. C. Rietjens.

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
