## [Editor Report · Decision Letter 0]

1 Mar 2020

Dear Dr Korfage, 

Thank you for submitting your manuscript entitled "Advance care planning in patients with advanced cancer; a six country, cluster-randomized clinical trial" for consideration by PLOS Medicine.

Your manuscript has now been evaluated by the PLOS Medicine editorial staff and I am writing to let you know that we would like to send your submission out for external peer review.

Kind regards,

Helen Howard, for Clare Stone PhD 

Acting Editor-in-Chief

PLOS Medicine 

plosmedicine.org

---

## [Decision Letter · Decision Letter 1]

4 Jun 2020

Dear Dr. Korfage,

Thank you very much for submitting your manuscript "Advance care planning in patients with advanced cancer; a six country, cluster-randomized clinical trial" (PMEDICINE-D-20-00632R1) for consideration at PLOS Medicine. 

[LINK]

In light of these reviews, I am afraid that we will not be able to accept the manuscript for publication in the journal in its current form, but we would like to consider a revised version that addresses the reviewers' and editors' comments. Obviously we cannot make any decision about publication until we have seen the revised manuscript and your response, and we plan to seek re-review by one or more of the reviewers. 

We expect to receive your revised manuscript by Jun 25 2020 11:59PM. Please email us (plosmedicine@plos.org) if you have any questions or concerns.

We look forward to receiving your revised manuscript. 

Sincerely,

Emma Veitch, PhD

PLOS Medicine

On behalf of Clare Stone, PhD, Acting Chief Editor,

PLOS Medicine

plosmedicine.org

*Please structure your abstract using the PLOS Medicine headings (Background, Methods and Findings, Conclusions) - "methods and findings" is a single subsection.

*At this stage, we ask that you include a short, non-technical Author Summary of your research to make findings accessible to a wide audience that includes both scientists and non-scientists. The Author Summary should immediately follow the Abstract in your revised manuscript. This text is subject to editorial change and should be distinct from the scientific abstract. Please see our author guidelines for more information: https://journals.plos.org/plosmedicine/s/revising-your-manuscript#loc-author-summary

*In the Abstract, it would be good to include a (short) statement of what the provision was for the Control arm, so the distinction between intervention and control can be understood.

*In the last sentence of the Abstract Methods and Findings section, please include a brief summary of any of the key limitation(s) of the study's methodology.

*It would be good to update the intext referencing callout at this stage if you can - please change to sequential numbers in square brackets (eg, [1, 2] etc) rather than superscript. If using referencing software this should be quick and easy.

*It would be important to robustly address the concerns raised by reviewer 3, particularly as this reviewer raises a number of points regarding the reliability of the analyses and whether some of the major conclusions drawn in the paper are indeed driven by the data. 

Comments from the reviewers:

Reviewer #1: Alex McConnachie

The following is a statistical review of Korfage et al's report on ACTION, an international, multicentre, cluster randomised trial of the use of advanced care planning for patients with advanced lung and colorectal cancer. Overall, I found the paper to be interesting, and well written. I do have a few comments, but these are generally quite minor.

The cluster randomised trial design is fully appropriate for this intervention, due to the likelihood of contamination. The problem is that participants appear to have known which arm of the trial they were being recruited into, given the very different consent rates (29 vs. 56%) in the two arms. Potentially, two quite different groups of patients may have been recruited into the trial. According to Table 1, the differences do not seem too extreme, plus the analysis includes plenty of adjustments, but with such different recruitment rates, there will always be the doubt that something could be biasing the overall results.

Had participants been asked for consent, and provided baseline measurements, before being told whether their hospital was in the ACP or control arm, this might have allowed some assessment of this differential engagement with the trial. Ideally, one would ask whether the factors associated with participation are the same in ACP hospitals and control hospitals. Is there any information about those who did not take part that could be used in this way?

I was able to replicate the sample size calculation from the information provided, but given that the detectable between-group difference was given a half a standard deviation, it is not clear why the required sample size should reduce when it was decided to change to a different primary outcome.

The analyses are perfectly acceptable. However, in the title of Table 2, the term "causal intercept" was not one I have encountered before (and an internet search did not shed any light). This may need some additional explanation. Another point on reading Table 2, is that it is not obvious whether positive and negative changes for each outcome represent an improvement or a deterioration. Also, though the presentation of the results is generally good, the ICCs for the different outcomes are not reported, as recommended by CONSORT.

Reviewer #2: Korfage and colleagues report the results of a large cluster-randomised controlled trial to examine the effect of implementing an ACP program in hospitalised patients with advanced cancer in six European countries. Intervention and control group showed no difference with regard to quality of life, the primary outcome, nor with most of the secondary outcomes including hospitalisation rate. Also in the intervention group, very few (even though more than in the control group) of the patients had an AD in their hospital charts.

Congratulation to the authors for this important paper that to me seems methodologically sound, and well written. I have only few and minor suggestions for the sections background, methods and results (see below). I have, however, a major concern that relates to the interpretation of the „negative" findings in the discussion section: 

The authors write: „The limited effectiveness of the complex ACP intervention evaluated in this study may be explained by 1) characteristics of the intervention, 2) patients' preferences regarding ACP and Ads or 3) the European context."

In fact there are a number of further possible explanations why this study showed no effect, and given the high attention this large and thoroughly done study is likely to attract, and the possible large impact of reporting a negative outcome in this relatively new and highly sensitive field, it is certainly warranted that these explanations are discussed in detail. In particular, a negative finding of a methodologically sound trial examining the effect of an instrument that has so strong merits as ACP may well cause the authors to take a step back and reconsider their assumptions and design. In my eyes, other passages in the manuscript can and should be shortened (for example the first 8 lines of the background) in order to allow for a thorough critical appraisal of the reasons why this study may have shown no effects. Here is my suggestion of the reasons that ought to be considered:

A. Reasons related to the intervention

1. The authors correctly report that they could not ensure intervention delivery. It would be worthwhile to make this relevant consideration become somewhat more prominent, for example by repeating the corresponding key results. 

a. First of all, it seems that conversations as the core of the intervention did take place most of the time (90%), occurred according to protocol (86%), they were extensive in time (93 minutes), contrary to most recommendations, though, they comprised rarely more than one encounter (1.3). 

b. However, only 41% of the patients who had ACP conversations proceeded to signing an advance care plan. This is very unusual. Do the authors have any hints from the process evaluation why this was the case? It would be helpful to understand why so many patients did not sign written advance care plans in order to appreciate how much of the intervention was in fact delivered. If it turns out that for practical or cultural reasons, the intervention was insufficiently realised, then this is of central importance for the interpretation of the results and needs to find entry also in the conclusion.

2. There is another important influential factor that warrants discussion. Implementing an ACP program such as Respecting Choices should not be limited to establishing a conversation process between facilitator, patient, and significant others; besides this personal level, there is also an institutional and regional level that logically must be considered if the ultimate goal of ACP that patients' preferences are known and honored is to be achieved. 

In particular, Respecting Choices is known to have always pushed both levels, the „systems change" level (institutional / organisational and regional development), and the personal conversation level. The ACTION trial has limited the intervention on the mere personal level. This needs to be discussed here: Why was the decision made? What were the personal, institutional and regional barriers observed in the respective countries? Are there any further indications that a „systems change" would have been necessary to let the intervention develop a positive impact on the clinical encounters?

B. Reasons related to the Study Design

If a study so well done as this one, and testing such an established intervention, does not find the expected group difference, it is warranted to reconsider whether the expectation was legitimate in the first place, for example asking the following questions:

1. Does improvement of quality of life belong to the goals that ACP is intended to reach? The authors correctly state that the ultimate goal of ACP is that patients' preferences are known and honoured when medical decisions have to be made. While I appreciate that relying on the medical team to know my well reflected treatment preferences for such a case may cause a certain sense of reassurance, at least immediately after the planning, personally I am having a hard time to understand how ACP could possibly exert a long-term effect on quality of life. Also, I am not aware of any strong evidence that this is likely to happen. I suggest, therefore, for the authors to discuss whether - in retrospect - the chosen primary (and related secondary) outcomes could really be expected to be affected by ACP even if optimally delivered and implemented.

2. But also with regard to clinical outcomes like hospitalisation: Is ACP likely to change the course of patients with advanced, non-curable cancer? Advance Care Planning, as the authors rightly quote from the EAPC White Paper, allows individuals to reflect, to plan and to document their preferences for future critical treatment decisions that may need to be taken when the now planning individual has become incapable of decision making. Any clinical direct effect of ACP, then, is to be expected (only) if patients are rendered incompetent at some time in the course of their illness AND if at that time medical decisions have to be taken for which the preferences planned and documented in advance can be regarded relevant. In advanced cancer, however, it would not be unexpected that most treatment decisions, even when approaching the very end of life, are made with the competent patient, while these patients become incompetent typically (i.e., with few exceptions) only late in the course of their dying process when any further attempts to sustain life would be futile and therefore medically not indicated (justifiable). Therefore, it would be interesting to learn whether the authors have collected any quantitative or qualitative impressions on whether medical treatment decisions had to be taken at all with study patients incapable of decision making so that an advance care plan, if present, could have become relevant. In any case, it ought to be discussed whether ACP is likely to make an effect in this patient group - unless there is an assumption that ACP indirectly also transforms care planning, i.e. the medical decision making between oncologists and competent, actively participating patients. Such an assumption, however, should also be made transparent in the discussion.

C. „The European context"

Given the strong case that the intervention may not have reached the actual decision making process (as indicated by a low rate of written advance care plans, and by the omisson of any institutional and regional implementation efforts in the sense of a probably necessary change management), and / or that the intervention in its nature may not be suitable to change many of the chosen outcomes, or unfold a relevant effect for the chosen patient population even if perfectly delivered and optimally measured, I would advise to reconsider the „European" argument. Here, the authors generalise their findings to ACP in general, suggesting that in Europe (i.e., for all patient populations in all European countries), there may be a lesser need for ACP because of a lesser rate of over-treatment, compared with the U.S. This is a far-reaching interpretation that in my perception is not warranted by the design nor by the results of this study. Given the alternative good possible explanations reflected above for the negative results found in this study, and the strong body of evidence for both overdiagnosis and overtreatment (and lack of patient-centred care) also in many European countries, this paper does not seem the right place to speculate whether ACP in general may not be a necessary or useful tool for European patients altogether.

Finally, I would suggest that the conclusions do reflect this critial discussion of the study results.

MINOR COMMENTS

Background

I may be wrong but I don't think I understand inhowfar the first 8.5 lines provide a background for this study given that ACP, as defined also by the authors in the following lines, is not an instrument to improve palliative care. 

METHODS

The intervention, first paragraph, line 4: What does the number in brackets (14) stand for? Reference # 14 seems not pertinent.

RESULTS

Intervention delivery: 

Given the low increase of written advance care plans in hospital charts, it would be interesting to add how many of the 396 patients who had ACP conversations came out of these conversations with written advance care plans. On page 8 (second paragraph) there is an information that „in the intervention group, 147 patients provided their facilitators with a copy of their completed MP forms." Since written advance care plans were one of three intervention components (#2), I would expect them to be reported here, as plain number and proportion.

In order to further help appreciate why hospital admission was not reduced by the intervention, it would be interesting to know how many patients opted for comfort-focused care in section D of their MP form. Were the MP forms analysed by preference?

Primary and secondary outcome measures:

I don't think the reference to the MICE procedure needs to be repeated here.

Reviewer #3: This is an interesting manuscript about a randomized control trial of advance care planning in patients with advanced cancer. The study's strengths are that it is a large study with 23 sites and is well designed with a cluster-randomized trial. However, my enthusiasm was dampened by several issues. I have the following comments and suggestions: 

1. In the background, the authors state that studies of ACP in patients with cancer are scarce. This is not a true statement. There are >1000 pubmed citations with "Advance Care Planning" and "Cancer". Furthermore, the authors cite several papers in their introduction. Please remove or edit this statement. 

2. I am concerned about the conclusion that "Alternate approached to support patient-centered end-of-life care in the population are needed". When I think about the causal pathway, ACP is most likely to impact decisions at true end-of-life. These patients were not necessarily followed to death and there are not outcomes about the location of death or quality of decision-making at true end of life. The outcomes in this study were not impacted, so the statement should be specific to those outcomes and not all encompassing end-of-life care, which may have been measured in other ways closer to actual end of life. 

3. I have a significant problem with the statement that "When cure is no longer possible, a focus on prolonging life is associated with worse quality of life." When treating metastatic, incurable cancer, we are most often treating with medications that are intended to BOTH prolong life and improve quality of life. While there are times when giving additional chemotherapy at end-of-life is both unlikely to prolong life or to improve quality, this statement is inaccurate the way it is written. 

4. Please define the abbreviation ACP in the first use in the introduction. 

5. Please clarify who was approached, who delivered the intervention. 

6. I am not clear on what the authors mean by the statement that the RC instructors followed the RC First and Advanced Steps training programme together. Please clarify. 

7. Please provide more detail on how many conversations were audio-recorded and were these practice conversations vs. conversations with patients with cancer. It would be important to know how the fidelity checklist was used. Did the facilitator have to get all on a certain number of times? This needs much more detail. 

8. I would like to see more information on the missingness of the data to ascertain if the multiple imputation is appropriate. Please provider a supplemental with the characteristics of those who did not complete the surveys, as this is common as people approach end of life and may impact interpretation of results. Also, please do a sensitivity analysis with a complete case. 

9. There is a large number of people who did not consent. This should be discussed. Also, the recruitment rate for the intervention was much lower than for the control, which doesn't make sense to me in a randomized study. Please explain this finding.

10. It is interesting that the authors state that they did not expect survival differences and yet 44% of people in the intervention died and only 39% in the control group died. This should be evaluated further, and particular attention paid to how this would impact the survey results in terms of missing data and QoL trajectories given that patients typically have a decline in QoL before death. I am not convinced by the P values provided given the small sample size and a clinically meaningful difference. 

11. I am concerned about the population differences, in particular lung vs. colorectal cncer between groups. The higher lung cancer in the intervention could results in lower quality of life and the shorter survival in this group given that these diseases have very different prognosis. These groups should be analyzed separately given the lack of balance in the randomization. 

12. Please provide explanation of why 10% of the intervention did not receive the ACP. This seems high. 

13. There should be more detail on how death was incorporated into the analysis. 

14. I struggled with the outcomes selected for the study. Please provide some context as to why and how the authors anticipate that an advance directive conversation would be expected to impact quality of life and emotional functioning at 3 and 6 months if this is not associated with end of life. I recognize that the authors cannot change the data or timing now, but it would be very helpful to understand their thinking in the design and why this was designed in this way. It might even be helpful to provide a conceptual model of impact. I do understand the decision-making outcomes. 

15. If possible, provide an explanation of the reason why the one hospital enrolled no patients. 

16. Please provide information available of why 13 patient did not complete any questionnaires. 

17. Was there an element that was commonly missed in terms of the fidelity? What were the key domains evaluated?

18. The amount of imputed data should be reviewed by a statistician. 

19. It is interesting that the decision-making self-efficacy, information provided by nurses/doctors, and rating of care received all declined from baseline. This could warrant comment in discussion. 

20. I am very confused by the recruitment rate in the discussion. If the randomization is at a site level, the some received standard of care and it is unclear why they declined. Please provide some literature review of why this might have occurred and/or participation in these types of trials. 

21. When the authors discuss the limited effectiveness, they should include some other studies and references to put their findings into context. The discussion should include references where they are making statements about the countries policies as well to verify legitimacy. 

22. The discussion of Europe s. US is troubling given that these are strong statements without any literature backing them. 

23. I again think the conclusions of alternate approaches is not supported fully by this data, rather that the endpoints chosen are not likely at that time to be impacted by AD.

[LINK]

---

## [Decision Letter · Decision Letter 2]

7 Aug 2020

Dear Dr. Korfage,

Thank you very much for submitting your manuscript "Advance care planning in patients with advanced cancer; a six country, cluster-randomized clinical trial" (PMEDICINE-D-20-00632R2) for consideration at PLOS Medicine. 

[LINK]

In light of these reviews, I am afraid that we will not be able to accept the manuscript for publication in the journal in its current form, but we would like to consider a revised version that addresses the reviewers' and editors' comments. Obviously we cannot make any decision about publication until we have seen the revised manuscript and your response, and we plan to seek re-review by one or more of the reviewers. 

We expect to receive your revised manuscript by Aug 28 2020 11:59PM. Please email us (plosmedicine@plos.org) if you have any questions or concerns.

We look forward to receiving your revised manuscript. 

Sincerely,

Adya Misra, PhD

Senior Editor 

PLOS Medicine

plosmedicine.org

Please adapt the title to fit PLOS Medicine style and replace the “;” with “:” 

Is ACTION an acronym? Please introduce on first view. The same goes for EORTC

Please provide participant demographics 

Please temper conclusions by including “our results show” or similar

Please can you provide 95% CI along with p values as needed, when quantifying the main results

The data statement needs to be modified as authors cannot be sole contacts for data requests as per PLOS data policy. Please provide institutional or research ethics committee details for data requests.

The author summary needs to be in bullet points 

References must be in square brackets and bibliography in Vancouver style please

Methods

Please include your trial registry early in the methods section

Please can you fill the CONSORT reporting guideline and provide this as Supplementary information. Early in the methods section please state that the study has been reported according to CONSORT. Please do not use page numbers when completing the checklist.

Please specifically provide details of informed consent and how it was received from participants

Discussion

Some sections require revision, in line with comments from Ref 2. There are several study design limitations that could have led to the results and therefore it would be cautious to avoid sentences such as “Fourthly, there might be relatively little to gain from ACP in Europe”. 

Please add a strengths and limitations section, to discuss the various limitations of your work that could have influenced the findings 

The role of the funder and conflicts of interest should be reported in the article meta-data only and removed from the main manuscript document

Specifically, some of the comments in Reviewer 2’s notes are pertinent and require significant revision to your discussion section: “What I am saying is: From the perspective that ACP requires institutional and regional implementation, the ACTION trial’s intervention was never likely to become effective because it lacked two out of three essential components. I understand that for the authors this is a difficult position to report, and the authors do not have to share this view, but since they cooperated with Respecting Choices, and report a negative result, in my eyes it is inevitable to make this possible severe objection clear and transparent in their discussion. Limiting the objection to „countries not familiar with ACP“, in contrast, not only misses but obscures the point”.

“That ACP may prepare for discussions of oncologic treatment is an interesting hypothesis that has no evidence base for this special patient population, certainly not in the paper of Rebecca Sudore et al. quoted by the authors (reference #10). 

In my eyes it should be discussed in the limitation section that the ACTION trial may have ended with a negative result because ACP may not be effective in changing quality of life and treatment decisions in the population of patients with advanced cancer since these rarely require treatment decisions to be based on advance care planning”.

“In the light of what can be said about possible shortcomings of the study design (including intervention and target population), this is not a „hypothesis that warrants further study“ but a hypothesis that does not warrant to be made here”

Comments from the reviewers:

Reviewer #1: Alex McConnachie, Statistical Review

I thank the authors for considering my original comments. In general, I am happy with their responses, and have no major concerns.

One comment on the sample size reduction, though. The original calculation was based on detecting a difference of half a standard deviation, or an effect size of 0.5. What is that on the scale of the EF4? Given the association between EF4 and EF10, what would be an "equivalent" difference be for the EF10? Then, given the SD of the EF10, what effect size would that equate to?

Also, it is notable that the ICC observed for the EF10 was smaller than was allowed for at the design stage. This is a good thing, as it should give more power, but does not seem to have been commented on.

Reviewer #2: See enclosure.

Reviewer #3: Thank you for addressing the comments, well done.

[LINK]

---

## [Editor Report · Decision Letter 3]

24 Sep 2020

Dear Dr. Korfage,

Thank you very much for re-submitting your manuscript "Advance care planning in patients with advanced cancer: a six country, cluster-randomized clinical trial" (PMEDICINE-D-20-00632R3) for review by PLOS Medicine.

I have discussed the paper with my colleagues and the academic editor and it was also seen again by xxx reviewers. I am pleased to say that provided the remaining editorial and production issues are dealt with we are planning to accept the paper for publication in the journal.

[LINK]

We look forward to receiving the revised manuscript by Oct 01 2020 11:59PM. 

Sincerely,

Adya Misra, PhD

Senior Editor 

PLOS Medicine

plosmedicine.org

Requests from Editors:

Please provide brief participant demographics such as sex, age range etc

Author summary- please remove the phrase “new element”

Page 5 “All competent patients…” do you mean competent to give consent or eligible for study participation? Please clarify

Page 6 “AD” should be introduced as advance directive (?) or other on first view 

Page 7 ACTION RC – please define RC on first view

Are the information leaflets, piloted checklists, surveys used in the intervention previously published anywhere? Please provide in text citations or copies as SI files as needed. 

Discussion- please add a sentence or two at the start describing what was done 

Discussion- Please introduce ICC in text, on first view 

Page 16 last paragraph- please change reference 6 to square brackets

Page 17 please include the wordpress link in the bibliography and add a citation, as needed

Please use Vancouver style for the references

Within the CONSORT checklist, please provide details of paragraphs and sections where the information has been provided. 

Please call fig 1 "Participant flowchart" and move it to the start of the "results"

CONSORT discourages statistical comparisons of baseline groups (as in table 1). Please remove these. 

Please remove "which may to some extent explain the negative results" from the limitations paragraph in the discussion

Please remove the spaces in the square brackets 

Please provide full access details for some of the references, for example ref 9, 16 etc

Please provide p values up to three decimal places only. There are some "p<0.0001" in the supplementary tables which should be revised. 

Comments from Reviewers:

[LINK]

---

## [Editor Report · Decision Letter 4]

19 Oct 2020

Dear Ms. Korfage, 

On behalf of my colleagues and the academic editor, Dr. Gabrielle Rocque, I am delighted to inform you that your manuscript entitled "Advance care planning in patients with advanced cancer: a six country, cluster-randomized clinical trial" (PMEDICINE-D-20-00632R4) has been accepted for publication in PLOS Medicine. 

PRODUCTION PROCESS

Before publication you will see the copyedited word document (within 5 business days) and a PDF proof shortly after that. The copyeditor will be in touch shortly before sending you the copyedited Word document. We will make some revisions at copyediting stage to conform to our general style, and for clarification. When you receive this version you should check and revise it very carefully, including figures, tables, references, and supporting information, because corrections at the next stage (proofs) will be strictly limited to (1) errors in author names or affiliations, (2) errors of scientific fact that would cause misunderstandings to readers, and (3) printer's (introduced) errors. Please return the copyedited file within 2 business days in order to ensure timely delivery of the PDF proof. 

If you are likely to be away when either this document or the proof is sent, please ensure we have contact information of a second person, as we will need you to respond quickly at each point. Given the disruptions resulting from the ongoing COVID-19 pandemic, there may be delays in the production process. We apologise in advance for any inconvenience caused and will do our best to minimize impact as far as possible.

PRESS

PROFILE INFORMATION

Thank you again for submitting the manuscript to PLOS Medicine. We look forward to publishing it. 

Best wishes, 

Adya Misra, PhD

Senior Editor 

PLOS Medicine

plosmedicine.org